# High-resolution genomic analysis to investigate the impact of the invasive brushtail possum (*Trichosurus vulpecula*) and other wildlife on microbial water quality assessments

Marie Moinet[1¤]*, Lynn Rogers[1], Patrick Biggs[2,3], Jonathan Marshall[4], Richard Muirhead[5], Megan Devane[6], Rebecca Stott[7], Adrian Cookson[1,2]*

1 Hopkirk Research Institute, AgResearch, Palmerston North, New Zealand, 2 mEpiLab, School of Veterinary Science, Massey University, Palmerston North, New Zealand, 3 School of Natural Sciences, Massey University, Palmerston North, New Zealand, 4 School of Mathematical and Computational Sciences, Massey University, Palmerston North, New Zealand, 5 Invermay Agricultural Centre, AgResearch, Mosgiel, New Zealand, 6 Institute of Environmental Science and Research Ltd. (ESR), Christchurch, New Zealand, 7 National Institute of Water and Atmospheric Research (NIWA), Hamilton, New Zealand

¤ Current address: Institute of Environmental Science and Research Ltd. (ESR), Porirua, New Zealand
* Adrian.Cookson@AgResearch.co.nz (AC); Marie.Moinet@gmail.com (MM)

**Data Availability Statement:** WGS data of the 101 bacterial isolates and gnd metabarcoding

## Abstract

*Escherichia coli* are routine indicators of fecal contamination in water quality assessments. Contrary to livestock and human activities, brushtail possums (*Trichosurus vulpecula*), common invasive marsupials in Aotearoa/New Zealand, have not been thoroughly studied as a source of fecal contamination in freshwater. To investigate their potential role, *Escherichia* spp. isolates (n = 420) were recovered from possum gut contents and feces and were compared to those from water, soil, sediment, and periphyton samples, and from birds and other introduced mammals collected within the Mākirikiri Reserve, Dannevirke. Isolates were characterized using *E. coli*-specific real-time PCR targeting the *uid*A gene, Sanger sequencing of a partial *gnd* PCR product to generate a *gnd* sequence type (gST), and for 101 isolates, whole genome sequencing. *Escherichia* populations from 106 animal and environmental sample enrichments were analyzed using *gnd* metabarcoding. The alpha diversity of *Escherichia* gSTs was significantly lower in possums and animals compared with aquatic environmental samples, and some gSTs were shared between sample types, e.g., gST535 (in 85% of samples) and gST258 (71%). Forty percent of isolates *gnd*-typed and 75% of reads obtained by metabarcoding had gSTs shared between possums, other animals, and the environment. Core-genome single nucleotide polymorphism (SNP) analysis showed limited variation between several animal and environmental isolates (<10 SNPs). Our data show at an unprecedented scale that *Escherichia* clones are shared between possums, other wildlife, water, and the wider environment. These findings support the potential role of possums as contributors to fecal contamination in Aotearoa/New Zealand freshwater. Our study deepens the current knowledge of *Escherichia* populations in under-sampled wildlife. It presents a successful application of high-resolution genomic

sequences of the 106 animal and environmental samples have been deposited to the NCBI under BioProject number PRJNA987865 (BioSamples SAMN35994385 to 35994485 for WGS and SAMN36426751 to 36426856 for metabarcoding) in the form of forward and reverse raw reads fastq files.

**Funding:** This work was funded by the New Zealand Ministry for Business, Innovation and Employment's Our Land and Water (Toitū te Whenua, Toiora te Wai) National Science Challenge, contract C10X1901, as part of the 'Faecal source tracking' program (Adrian Cookson). The salary of Marie Moinet was funded through the Crown Research Institutes' Strategic Science Investment Fund as part of the 'Food Integrity' project. The funders had no role in study design, data collection and interpretation, or the decision to submit the work for publication.

**Competing interests:** The authors have declared that no competing interests exist.

methods for fecal source tracking, thereby broadening the analytical toolbox available to water quality managers. Phylogenetic analysis of isolates and profiling of *Escherichia* populations provided useful information on the source(s) of fecal contamination and suggest that comprehensive invasive species management strategies may assist in restoring not only ecosystem health but also water health where microbial water quality is compromised.

## Introduction

*Escherichia coli* is a common inhabitant of the gut microbiome of warm-blooded terrestrial mammals and birds. The presence of *E. coli* in waterways suggests that fecal contamination has occurred, and *E. coli* are therefore widely used by regulatory authorities as fecal indicator bacteria (FIB) for routine monitoring of water quality utilizing frameworks which provide human health risk assessments for safe drinking and recreational use. Where water quality is compromised, identifying the likely source(s) of contamination is important when targeting effective mitigation strategies to improve water quality, and fecal source tracking tools have been developed, with a focus on livestock, human sources and activities, and birds [1–4]. Chemical and genetic markers have also been developed to detect gull and wildfowl fecal contamination in water [3] or differentiate human from possum sources [5, 6]. However, these tools often require large sample volumes, perform better when there is a dominant source of fecal pollution, and fail to provide unequivocal evidence of microbial sources [2, 7]. In contrast, Next Generation Sequencing appears as a promising tool for extending the fecal source toolbox beyond the current markers and targeted species [2].

Despite a high diversity of *E. coli* strains in wildlife [8], they have remained understudied worldwide [9] and few studies have investigated *E. coli* present in New Zealand wildlife [10]. None have undertaken a detailed characterization of *E. coli* strains found in invasive mammals [11–14]. In particular, the omnivorous brushtail possum (*Trichosurus vulpecula*), is a nationally significant invasive species throughout the country, at high densities (3-25/ha) in native broadleaf-podocarp forests [15] but also present in diverse habitats such as grasslands and urban habitats [16, 17]. This arboreal marsupial is known to consume endangered native species of insects, bird eggs and nestlings, and a significant amount of native plant foliage [18] leading to a national direct annual biomass loss of 1.9–3.82 million tons of $CO_2$ equivalent per year [19]. Considerable resources are spent in an attempt to control possum numbers not only due to their significant impact on endemic species and native biodiversity, but also as they may serve as spillover and spillback hosts of *Mycobacterium bovis* for dairy cattle [20]. Numbering tens of millions across the country [21], and known to harbor *E. coli* [22], the elimination of possums from the environment may positively impact the fecal microbiological quality of waterways as well as enhance biodiversity and endemic ecosystems.

Microbial water quality assessments can also be potentially compromised by the presence of *E. coli*-like or 'cryptic clades' of *Escherichia* [23]. These cryptic clades cannot be distinguished from fecal *E. coli* using conventional, culture-based water quality monitoring tests [24]. To date, eight cryptic clades of *Escherichia* (I-VIII) have been described [24–26]. Clade V was later designated as the novel species *E. marmotae* [27], clades III and IV have been defined as subspecies of *E. ruysiae* [28], and clade II as *E. whittamii* [29]. While clade VII appears to be phylogenetically close to clade II [26], clades VI-VIII have not yet been formally described. Except for clade I, considered a subspecies of *E. coli*, these 'naturalized' *E. coli*-like bacteria show enhanced survival and persistence in soil, water and sediment, but are rarely found in fecal samples [30, 31]. The hypothesis of an environmental habitat has been suggested [31],

but cryptic clades have also been isolated from a variety of wild animals, are under-represented in current culture collections and sequence databases, and whether wildlife acts as a source or a sink for cryptic clades is still a subject of debate [32, 33]. The recent discovery of *E. marmotae* and *E. ruysiae* in the New Zealand environment [34] and their inability to be differentiated from fecal *E. coli* using standard culture-based methods has created uncertainty as to whether, in some instances, current water quality monitoring assessments may not provide an accurate indication of human health risk and whether subsequent mitigations are necessary to address *E. coli* exceedances [23]. Wildlife, as either a potential reservoir of, or amplifier of those cryptic clades, may also contribute to misinterpretation of water quality monitoring test results and indecision on the implementation of remedial actions or recommendations by water managers.

This study aimed to examine the profile of *Escherichia* populations in possum fecal specimens to determine whether this introduced predator species contributes fecal *E. coli* bacteria, as well as *E. coli*-like naturalized *Escherichia* species with the potential to impact water quality assessments. A dual approach was used: firstly, *E. coli* and *E. coli*-like naturalized *Escherichia* species isolated from possums were submitted for whole genome sequencing (WGS) and phylogenomic analysis; secondly, amplicon metabarcoding targeting *gnd*, a hypervariable gene encoding 6-phosphogluconate dehydrogenase, found in all *E. coli* and *E. coli*-like naturalized *Escherichia* species, was used to obtain *E. coli* community profiling directly from sample enrichments. Profiling was achieved by comparing *gnd* amplicons with gndDb, a custom database containing 644 distinct *gnd* sequence types (gSTs) found in *E. coli* and *E. coli*-like naturalized *Escherichia* species [35]. *E. coli* isolates and community profiles were then compared with those obtained from environmental samples and from the gut contents of other introduced mammals and feces of birds collected in the vicinity.

## Results

### Trapping and environmental sampling

Over the two-month sampling period within the Mākirikiri Reserve (Fig 1), there were 1,523.5 corrected trap-nights (successful traps, and traps sprung but empty were counted as set for half of the associated nights), and 3.8 captures per 100 trap-nights (C100TN). Timms traps were the most effective trap type used (6.2 C100TN), followed by Trapinators (3.7 C100TN) and GoodNature A24 $CO_2$-powered traps (1.3 C100TN). No captures were made with the GoodNature A12 traps. Overall, 49 possums, 5 ship rats, 3 hedgehogs and 1 ferret (all invasive species) were captured and sampled (Table 1, Details in S1 Table in S1 File). Over four environmental sampling visits, a total of eight samples of each environmental type (water, periphyton, sediment and soil) and 16 fecal samples (nine of avian origin, and seven from possums) were collected from the two environmental sampling points (Fig 1).

### *E. coli* enumeration

*E. coli* concentrations in the eight water samples assessed using the same routine water quality monitoring method (Colilert-18 and QuantiTray/2000) as the local authorities were all greater than 260 *E. coli* per 100 mL and according to local water quality risk assessment guidelines [36] were considered of potential poor quality for NZ freshwaters (S2 Table in S1 File). The geometric mean of the Most Probable Number (MPN) of *E. coli*/100 mL of water was 317 (geometric standard deviation factor ×/÷ 1.11) at the Mākirikiri sampling point and 487 (×/÷ 1.51) at the Confluence.

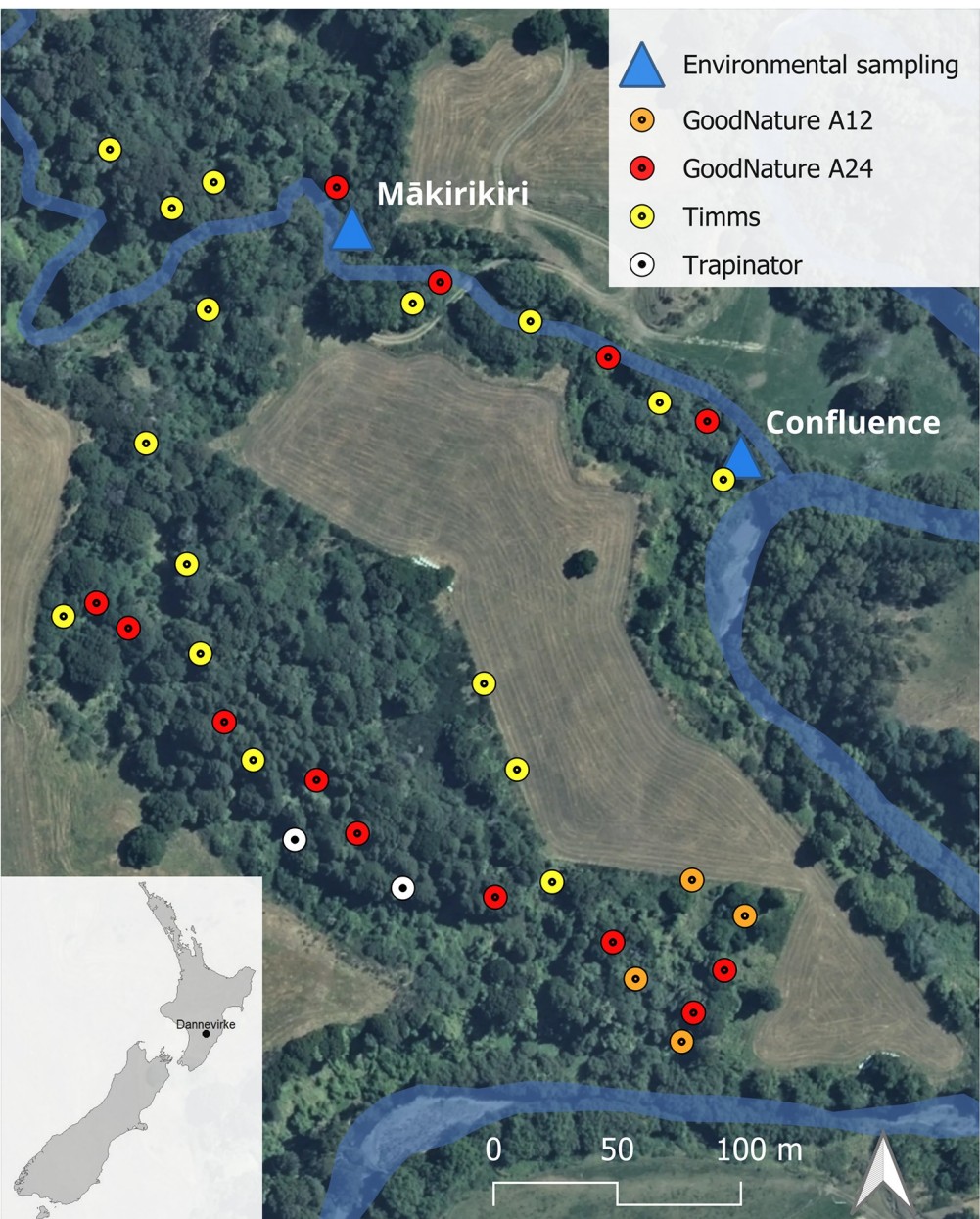

**Fig 1. Trap line within the Mākirikiri Reserve, Dannevirke, New Zealand, and two environmental sampling points, along the Mākirikiri Stream and just before the confluence with the Mangatera River (outlined in blue).** Background aerial imagery sourced from Toitū Te Whenua LINZ CC BY 4.0 Imagery Basemap contributors.

## *E. coli* isolate diversity

*uidA* **real-time PCR (RT-PCR) and confirmation of species.** A total of 420 isolates of presumptive *E. coli* were obtained from 105/106 samples (99.1%) following enrichment in EC broth and subculturing onto CHROMAgar ECC, an *E. coli* selective medium (Table 1, S1, S2 Tables in S1 File). To distinguish *E. coli* bacteria from *E. coli*-like bacteria (cryptic clades of *Escherichia*), an *E. coli* specific RT-PCR targeting the *uidA* gene was conducted and 389/420 isolates (92.6%) were provisionally identified as *E. coli*. The remaining 31 *uidA*-qPCR negative

**Table 1. Information at the isolate and sample level stratified by sample type.** At the isolate level, number obtained, *gnd*-sequence typed (gST) and submitted to whole genome sequencing (WGS), number of different gSTs and phylogroups detected, and at the sample level, number of reads of *gnd* sequences obtained by metabarcoding, percentage of the total number of reads, gamma and alpha diversity (i.e., total number of gSTs across samples and mean number of gSTs by sample).

| Sample type[1] | Number of isolates[2] | *gnd* typed[3] | Number of different gSTs detected | sent for WGS | Phylogroups[4] A | B1 | B2 | D | E | *E. m* | Number of samples | Sum of reads | % of total | Number of different gSTs detected[5] | mean number of gSTs[5] ±SD per sample | Number of 'cryptic gSTs' detected[5,6] | Samples with 'cryptic gSTs' (%)[5,6] |
|---|---|---|---|---|---|---|---|---|---|---|---|---|---|---|---|---|---|
| Possum (gut) | 196 | 91 | 16 | 58 | . | . | 36 | 19 | . | 3 | 49 | 1 086 269 | 32.8% | 58 | 4.1 ± 2.0 | 7 | 12 (24%) |
| Possum (faeces) | 28 | 14 | 8 | 5 | . | . | 5 | . | . | . | 7 | 183 934 | 5.6% | 45 | 9.4 ± 9.1 | 4 | 3 (43%) |
| Ship rat (gut) | 20 | 10 | 8 | 5 | 1 | 1 | 2 | . | . | 1 | 5 | 250 649 | 7.6% | 85 | 22.8 ± 17.5 | 11 | 3 (60%) |
| Hedgehog (gut) | 12 | 6 | 4 | 2 | . | 1 | 1 | . | . | . | 3 | 168 779 | 5.1% | 63 | 26.0 ± 6.0 | 3 | 2 (67%) |
| Ferret (gut) | 4 | 2 | 1 | 1 | . | . | 1 | . | . | . | 1 | 19 225 | 0.6% | 20 | 20 | 0 | 0 (0%) |
| Avian faeces | 36 | 20 | 13 | 8 | 1 | 1 | 3 | 2 | . | 1 | 9 | 335 309 | 10.1% | 129 | 21.2 ± 10.2 | 11 | 8 (89%) |
| *Total Possum* | *224* | *105* | *20* | *63* | . | . | *41* | *19* | . | *3* | *56* | *1 270 203* | *38.4%* | *83* | *4.8 ± 4.0* | *9* | *15 (27%)* |
| *Total Animal* | *296* | *143* | *36* | *79* | *2* | *3* | *48* | *21* | . | *5* | *74* | *2 044 165* | *61.8%* | *243* | *9.0 ± 10.0* | *15* | *28 (38%)* |
| Water | 32 | 17 | 11 | 6 | . | 3 | 1 | 1 | 1 | . | 8 | 391 466 | 11.8% | 267 | 87.5 ± 14.0 | 9 | 8 (100%) |
| Sediment | 32 | 17 | 16 | 5 | . | 1 | . | 2 | 2 | . | 8 | 248 259 | 7.5% | 207 | 55.6 ± 31.0 | 11 | 6 (75%) |
| Soil | 28 | 14 | 9 | 6 | . | . | 2 | . | . | 4 | 8 | 305 042 | 9.2% | 78 | 12.8 ± 6.3 | 8 | 5 (63%) |
| Periphyton | 32 | 16 | 15 | 5 | 1 | 1 | 1 | . | . | 2 | 8 | 318 548 | 9.6% | 299 | 131.3 ± 35.1 | 8 | 8 (100%) |
| *Total environment* | *124* | *64* | *41* | *22* | *1* | *5* | *4* | *3* | *3* | *6* | *32* | *1 263 315* | *38.2%* | *426* | *71.8 ± 49.9* | *16* | *27 (84%)* |

[1] faeces were found in the environment, gut content was sampled in trapped animals during necropsy;

[2] four isolates per sample, except one soil sample for which no isolates were obtained;

[3] up to two isolates tested per sample;

[4] determined from WGS sequences using the *in silico* ClermonTyping tool;

[5] gSTs with less than 10 reads not included;

[6] gSTs described only in cryptic clades of *Escherichia* (*E. marmotae*, *E. ruysiae*, *E. whittamii*) in the gndDb database.

isolates were classed as putative cryptic clades. Of these, 13 came from soil (from four different samples), 9 from gut contents (from three different possums and a rat), 5 from periphyton (from two different samples) and 4 from the same avian fecal sample.

**Gnd sequence typing of *Escherichia* isolates.** Typically, two isolates per sample (207 isolates in total) were further characterized using *gnd* PCR amplification and Sanger sequencing to determine their respective gSTs (Table 1). When possible, isolates with contrasting *uidA* RT-PCR Cq were selected for PCR and subsequent Sanger sequencing to maximize the diversity of isolates typed. The overall gamma diversity (i.e., the total number of different gSTs across the 207 isolates) was 63. Out of the 105 isolates from possum feces and gut contents that underwent *gnd* amplification and Sanger sequencing, 20 different gSTs were assigned (Table 1). The most frequent were gST535 and gST258 (29 isolates each, in total 55% of fecal

and gut possum isolates *gnd*-typed). gST258 was also the most frequent gST in isolates from other animals (9/38 isolates *gnd*-typed, 24%). Out of the 64 isolates from environmental samples, 41 different gSTs were assigned. The three most frequent gSTs were gST152 (five isolates, 7.8% of environmental isolates), gST535 and gST587 (both four isolates, each 6.3% of environmental isolates respectively). Forty percent of the isolates had a gST shared between possums, other animals, and the environment (S1A Fig in S2 File).

gST535 isolates were detected from 2 different water samples and 1 soil sample (at both sampling sites but on three different occasions), and 17 possum guts, 3 possum feces, and 1 ship rat (S3 Table in S1 File). We isolated gST258 from 22 different possum guts, 3 different avian fecal samples, 1 sediment, 1 ship rat, 1 hedgehog and 1 ferret, but interestingly gST258 was not isolated from possum feces.

Among the 31 putative *E. marmotae* isolates (*uidA* RT-PCR with Cq>35), 17 were typed and assigned with a gST from gndDb [ref. 35] and 7 different gSTs were assigned: gST541 (from one soil and two different possums gut contents), gST587 (from a soil and a periphyton sample), gST537 (from a periphyton and a ship rat gut content), gST540 and gST543 (both from soil samples), gST591 (from possum gut contents), and gST548 (from an avian fecal sample). Only one gST associated with *E. marmotae* was found per sample. These *E. marmotae* were isolated from 50% of soil (4/8), 25% of periphyton (2/8), 20% of rat gut contents (1/5), 11.1% of avian feces (1/9) and 6.1% of possum gut samples (3/49, Table 1).

**Whole genome sequencing and phylogenetic analyses of isolates.** WGS was undertaken on 101 isolates chosen to provide information on the potential role of mammalian pests on environmental contamination. Therefore, highly abundant *E. coli* gST258 and gST535 were selected for high resolution phylogenetic analysis, including 30 gST258 found in possum, ship rat, hedgehog, ferret, avian feces, and sediment samples, and 23 gST535 found in possum gut contents and feces, ship rat, soil and water samples. Eleven of the 31 putative cryptic *Escherichia* clade isolates (one per sample) found in possum, ship rat, avian feces, periphyton and soil samples and eight other *E. coli* gSTs found in several different sample types were also selected (S4 Table in S1 File).

*De novo* assembled genomes varied in size from 4.48 to 5.24 Mb (average 4.75 ± 0.222 Mb for 11 *E. marmotae* isolates and 5.07 ± 0.142 Mb for 90 *E. coli* isolates) (S5 Table in S1 File). The mash analysis confirmed the species groupings and separated the selected *E. coli* further into five different phylogroups: 3 in phylogroup A, 8 in B1, 52 in B2, 24 in D and 3 in E (Table 1). *In silico* MLST implementing the Achtman seven-locus MLST scheme, identified 30 different Sequence Types (STs), the most abundant equally being ST681 (n = 19, 18.8%, found in gST535 isolates) and ST714 (n = 19, 18.8%, found in gST258 isolates), and ST1170 (n = 9, 8.9%, found in gST371 isolates) (S4 Table in S1 File). Beside the ubiquitous bla$_{EC}$ β-lactam resistance gene, no antimicrobial resistance genes of public health concern were detected (S4 Table in S1 File). The *eae* gene encoding the intimin protein was detected in eight of the 91 *E. coli* genomes, belonging to two clonal groups: *eae* subtype α1 in three sediment, water, and avian feces genomes (phylogroup D O7:H6 ST362), and β2 in five possum genomes (phylogroup B2 O87:H6 ST3303, S4 Table in S1 File). All phylogroup B2 isolates had several genes associated with the extraintestinal pathogenic *E. coli* (ExPEC) pathotype (e.g., *chuA*, *fyuA*, *focC*, *irp2* and *sfaD* genes were detected in all gST535 isolates, S4 Table in S1 File).

The core SNP phylogeny of the 101 genomes selected used 343,828 SNPs (6.8% of the average *Escherichia* genome size of 5.04Mb) and confirmed the phylogroup classification (Fig 2). It revealed several clusters with a low number of SNPs (Fig 2). The coverage (i.e., the proportion of each genome covered in the SNP analysis) varied between 67.7% and 99.8% and was higher for *E. coli* genomes (84.6 ± 8.5%) than *E. marmotae* genomes (68.8 ± 0.7%). The average number of SNPs was 46,132 ± 24,914 among *E. coli* genomes, 6,935 ± 1,870 among *E. marmotae*

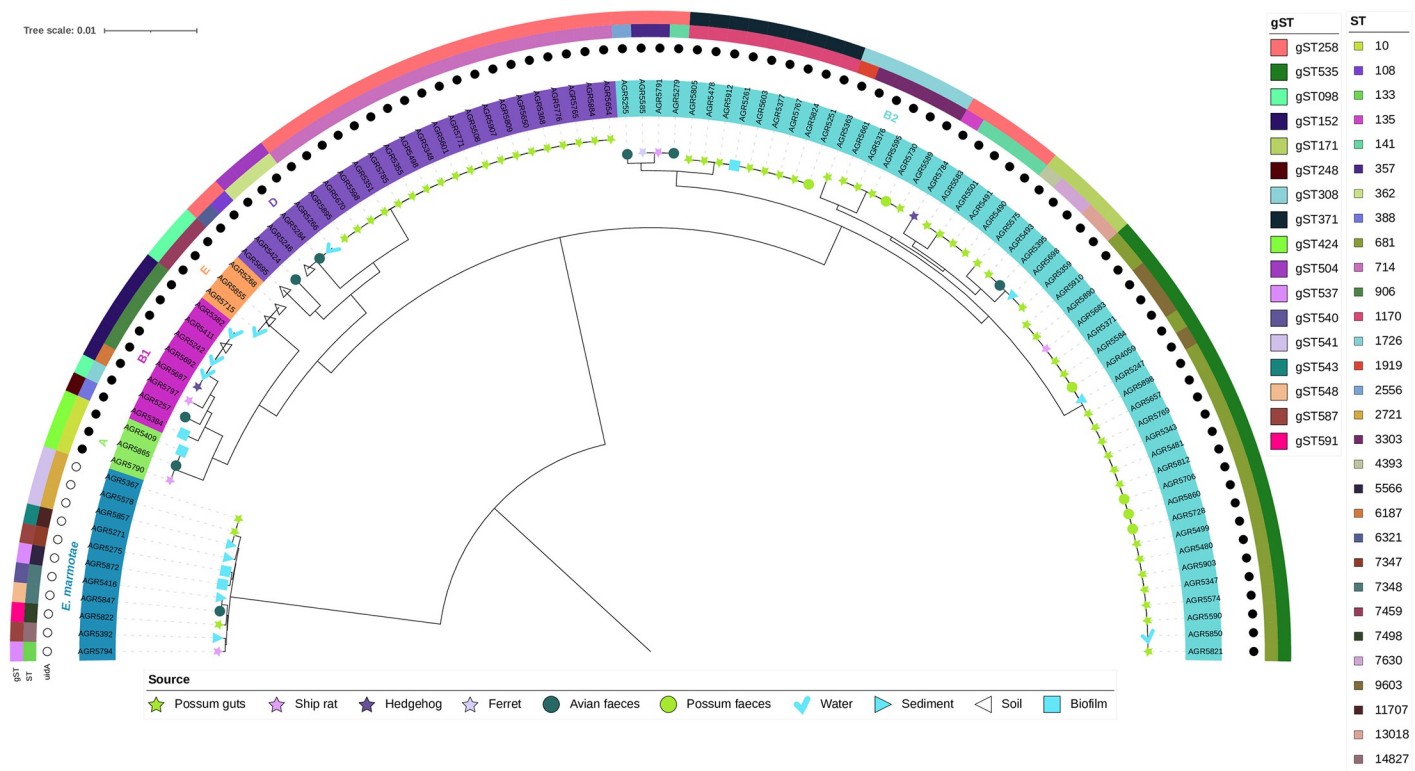

**Fig 2. Core SNP phylogeny of 101 *Escherichia* isolates from brushtail possum (*Trichosurus vulpecula*), other invasive mammals, birds and environmental samples from the Mākirikiri Reserve, Dannevirke, New Zealand.** Maximum likelihood tree rooted at midpoint of 90 isolates of *Escherichia coli* and 11 isolates of *E. marmotae* reconstructed using 343,828 core SNPs (6.82% of average *E. coli* genome size) with IQTree and edited using the iToL webserver. Isolate metadata is included for *gnd* Sequence Type (gST), Achtman MLST Sequence Type (ST), *uidA* gene detection by qPCR (black dot = positive) and isolation source. The tree includes AGR4059 [ref. 34], a gST535 isolated on 12 March 2018 from the same reserve and used as an internal reference sequence in the SNP analysis.

genomes, and 216,980 ± 932 SNPs between all 101 isolates which underwent WGS. Almost all gSTs selected were monophyletic (i.e., belonged to one phylogroup) with the notable exception of gST258 found in phylogroups B2 and D (Fig 2). Genomes within the gST535 cluster had between 0 and 101 SNPs (45 ± 37 SNPs, 96.2 to 99.8% coverage) and were re-examined with WGS data from other gST535 strains previously recovered from the same location [34] for further higher resolution core SNP analysis. The core SNP analysis of gST535 isolates revealed an extremely conserved cluster. The maximum distance between two isolates was 826 SNPs and most isolates were less than 10 SNPs different (Fig 3). Other notable clusters were gST371 (phylogroup B2, eight possums and a periphyton, 8 ± 9 SNPs, 83% coverage), gST424 (phylogroup A, one periphyton, one rat, one avian feces, 13 ± 4 SNPs, 76% coverage), gST171 (phylogroup B2, two possums, one avian feces, one soil, 21 ± 12 SNPs, 81% coverage), and gST541 (*E. marmotae*, two possums, one soil, 89 ± 6 SNPs, 68% coverage).

## *E. coli* community composition

Amplicon metabarcoding targeting *gnd* was performed on all samples to investigate *E. coli* populations at the sample level. After trimming and filtering, a total of 3,307,480 reads and 568 unique amplicon sequence variants (ASVs) were obtained from the 106 sample libraries of water, sediment, soil, periphyton, feces, and gut contents (Table 1). Just over half of the ASVs (297/568, 52%), representing 94.5% of the reads (3,126,509), could be assigned to existing gSTs

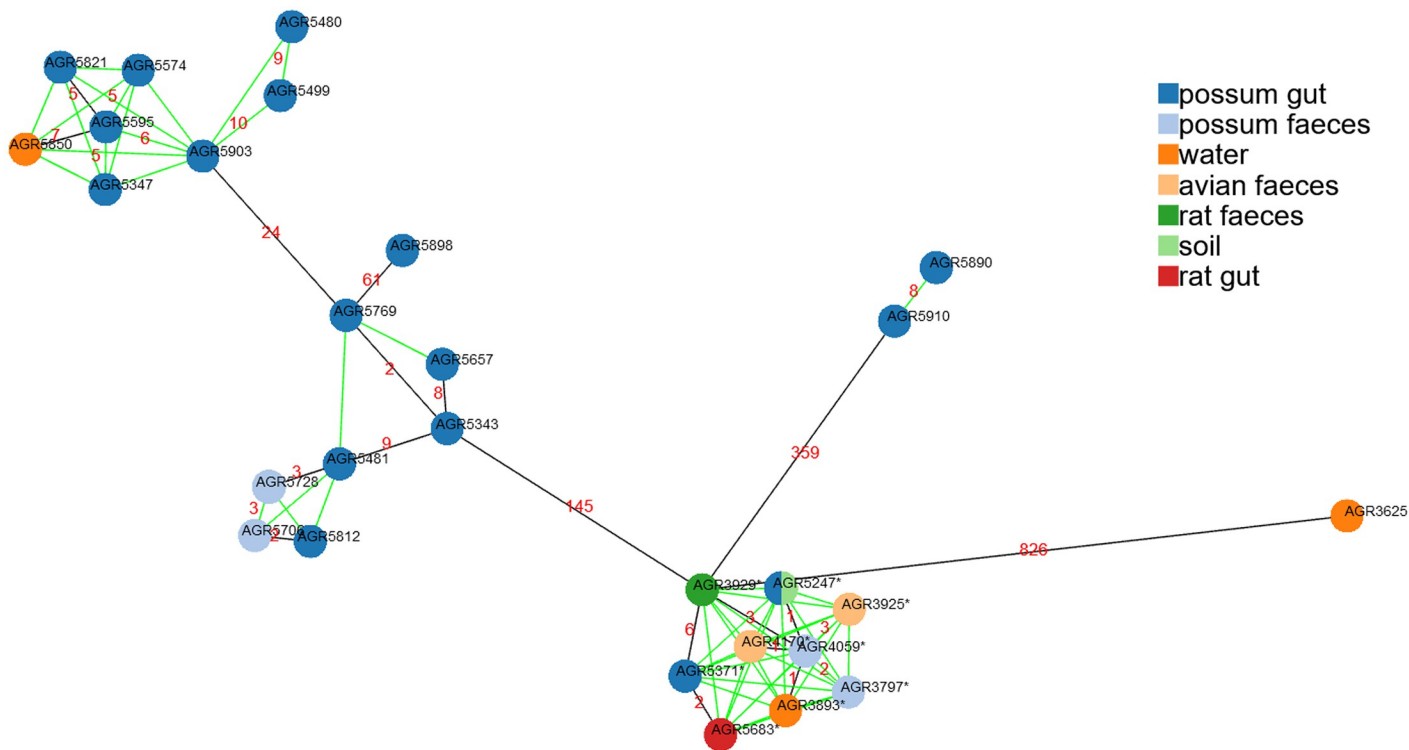

**Fig 3. Genomic relatedness between 29 *Escherichia coli* isolates *gnd* Sequence Type 535 (gST535, corresponding to Achtman MLST ST681 or ST11707 (the latter indicated with an \*)).** N locus variant graph of the core genome constructed with the goeBURST algorithm using PHYLOViZ Online based on 1,317 core single nucleotide polymorphisms (SNPs). Distances in number of SNPs between isolates are noted in red. AGR5584 (similar to AGR5247, i.e., 0 SNPs) and AGR5860 (similar to AGR5728) are masked. All isolates with distances equal or below 10 are linked with green lines. AGR3625 to AGR4170 (n = 7) were isolated between September 2017 and May 2018 [ref. 34], and AGR5247 to AGR5910 (n = 23) were isolated in November and December 2020 (this study).

already present in the gndDb database [35]. Of the top 100 ASVs by read abundance, 92% matched existing gSTs. A BLAST search done on the first 20 unknown ASVs did not identify any 100%-matching *gnd* alleles except in one *E. fergusonii* (Accession number CP057657), one *Citrobacter braakii* (CP069775), and one further *E. coli* (KY115228, subsequently added to the gndDb). The dada2 error model conserved 68 ASVs with ≤10 reads across all sample libraries that were kept for downstream analyses except when mentioned. Out of the 106 samples, reads obtained from the 56 possum samples represented 38.4% of the total and reads from all 74 animal sources represented 61.8% of the reads. Periphyton had the highest number of ASVs assigned on average, and mammal sources and soil the lowest (Tables 1, 2). All gSTs included in the mock community positive control libraries were detected in accordance with anticipated read numbers according to previous studies [37].

Numerous gSTs were shared by different sample types (Fig 4, S1B Fig in S2 File). Overall, gST535 and gST258 were the most frequently encountered gSTs in all types of samples (Table 3). The former gST was detected (with ≥10 reads) in 85% (90/106) of samples and the latter in 71% (75/106). A barplot showing the distribution of the 20 most abundant gSTs (in terms of number of reads) is available in S2 Fig in S2 File.

The observed richness was not influenced by the sequencing depth (i.e., total mapped reads per sample, S3 Fig in S2 File). Despite the higher number of reads obtained from possums compared to other sample types, the alpha diversity (Shannon index) estimated using the Div-Net approach was higher in periphyton, water and sediment than in animal sources including

**Table 2. Outputs of the breakaway (A) and DivNet (B) models.**

| A (richness) | Estimates | Standard Errors | p-values |
|---|---|---|---|
| (Intercept) | 92.9 | 2.1 | 0 |
| predictorsPeriphyton | 48.3 | 6.4 | 0 |
| predictorsSediment | -32.4 | 6.4 | 0 |
| predictorsSoil | -74.1 | 8.1 | 0 |
| predictorsPossum guts | -87.3 | 3.6 | 0 |
| predictorsPossum faeces | -60.9 | 18.1 | 0.001 |
| predictorsAvian faeces | -68.2 | 6.0 | 0 |
| predictorsHedgehog | -62.6 | 10.5 | 0 |
| predictorsShip rat | -67.1 | 9.1 | 0 |
| predictorsFerret | -66.9 | 18.1 | 0 |
| B (Shannon diversity index) | Estimates | Standard Errors | p-values |
| (Intercept) | 3.06 | 0.02 | 0.000 |
| predictorsPeriphyton | 0.28 | 0.08 | 0.001 |
| predictorsSediment | -0.83 | 0.19 | 0.000 |
| predictorsSoil | -0.14 | 0.52 | 0.789 |
| predictorsPossum guts | -2.00 | 0.02 | 0.000 |
| predictorsPossum faeces | -2.59 | 0.25 | 0.000 |
| predictorsAvian faeces | -0.45 | 0.14 | 0.001 |
| predictorsHedgehog | -0.61 | 0.15 | 0.000 |
| predictorsShip rat | 0.36 | 0.20 | 0.074 |
| predictorsFerret | -1.71 | 0.51 | 0.001 |

Water is the intercept and other sample type estimates are provided relative to the intercept, e.g., the estimated richness in Periphyton is 48.3 Amplicon Sequence Variants more than in Water (92.9 + 48.3 = 141.2).

possums. Indeed, the difference in the alpha diversity of water and other sample types was significant at the 0.05 level, except for soil (p = 0.789) and ship rats (p = 0.074, Table 2, S3 Fig in S2 File). As regards to the beta diversity, we failed to reject the null hypothesis of equal median measured relative abundance across sample types at the 0.05 level (bootstrap p-value = 0.714). In other words, despite lower alpha diversities in animal compared to environmental samples, we did not detect significant difference in measured beta diversity (Bray-Curtis distances) between sample types (S4 Fig in S2 File).

Out of 26 different gSTs described only in cryptic clades in the gndDb database [35], 17 were detected in the metabarcoding dataset, namely gST536, gST537, gST539 to gST546, gST548, gST587, gST591, gST593 (previously identified in isolates of *E. marmotae* in the gndDb), gST549, gST551 (previously identified in *E. ruysiae*), and gST614 (previously identified in *E. whittamii*). These 17 'cryptic gSTs' were found in all types of samples except the single ferret gut sample, with the highest diversity in sediment, avian feces, and ship rats (11 gSTs each, Table 1). The most abundant was gST587 (Table 3).

## Discussion

Wildlife is a potential source of fecal contamination in waterways [10], therefore, our hypothesis was that in areas where no pest management is undertaken, invasive mammalian species may be an important deleterious source of freshwater microbial contaminants. The contribution, however, of FIB in waterways from invasive species such as brushtail possums, had not been investigated in detail until now. The comparison of putative/presumptive *E. coli* isolates

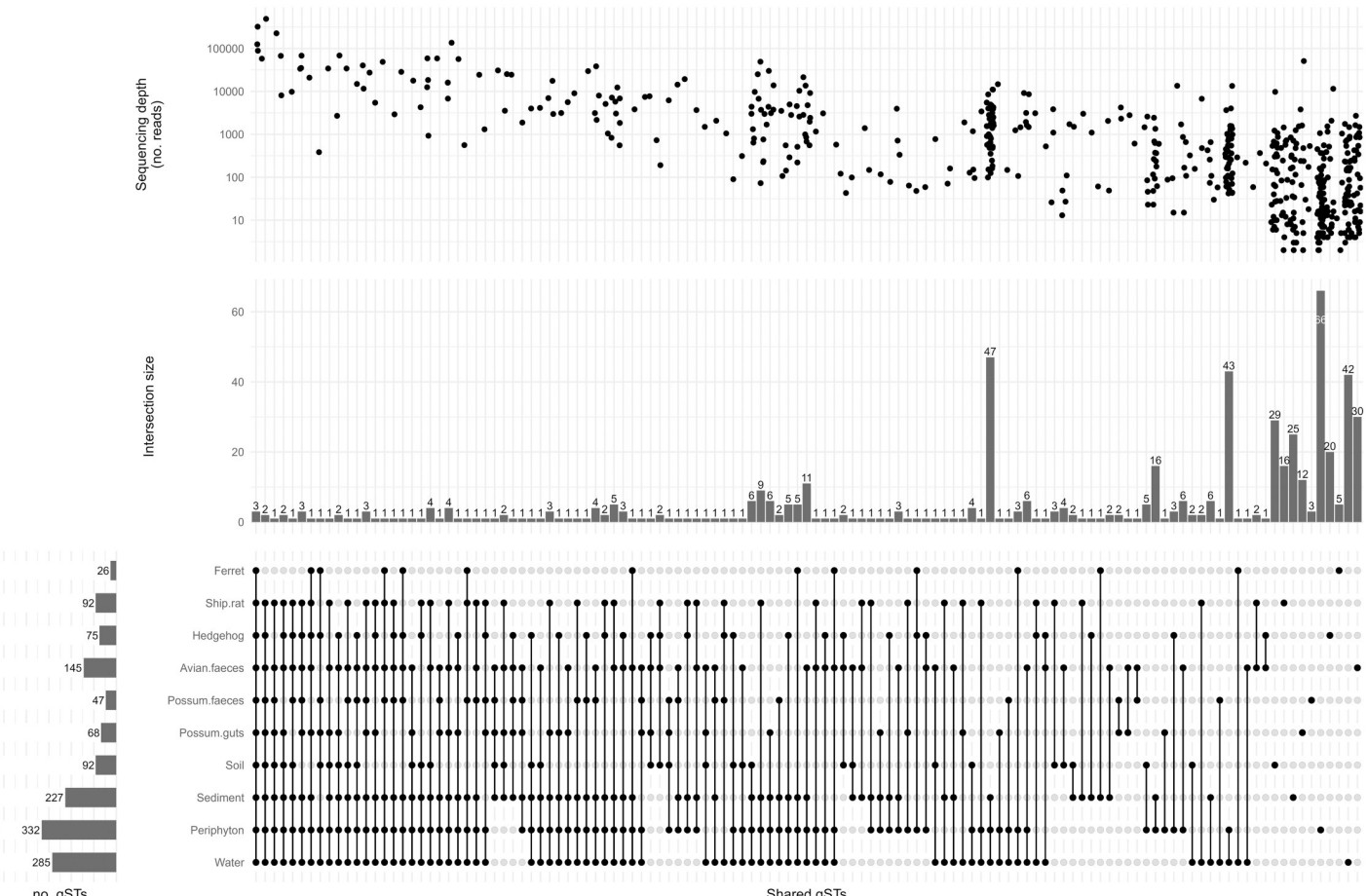

**Fig 4. Upset plot displaying the shared amplicon sequence variants or *gnd* Sequence Types (gSTs) between sets of sample types.** The black dots on the vertical intersection lines indicate the presence of shared gSTs between the corresponding sample types. Each horizontal bar gives the total of gSTs per sample type, while the vertical bars indicate the size of the intersections, i.e., the number of gSTs shared by combinations of sample types. The number of reads per gST associated with each intersection is indicated in the top scatterplot. The plot highlights the large overlap in *Escherichia* community composition across sample types, the high frequency of shared gSTs across the different sample types, and the higher relative abundance in terms of reads of shared gSTs.

**Table 3. Numbers of samples positive (%) for the 8 most common *gnd* Sequence Types (gST) detected by metabarcoding in animal (n = 74) and environmental (n = 32) samples.**

| Most frequent gSTs | in Animal samples | in Environmental samples | Overall |
|---|---|---|---|
| gST535 | 62 (84%) | 28 (88%) | 90 (85%) |
| gST258 | 55 (74%) | 20 (63%) | 75 (71%) |
| gST522 | 12 (16%) | 26 (81%) | 38 (36%) |
| gST152 | 11 (15%) | 22 (69%) | 33 (31%) |
| gST514 | 11 (15%) | 22 (69%) | 33 (31%) |
| gST308 | 14 (19%) | 17 (53%) | 31 (29%) |
| gST587 | 13 (18%) | 15 (47%) | 28 (26%) |
| gST231 | 5 (7%) | 22 (69%) | 27 (25%) |

Samples with 10 reads or less for a given gST were not included in the calculation.

and *Escherichia* community profiles obtained from possums, other wildlife, and environmental samples identified clonal strains of *E. coli* highly prevalent in both animal and environmental samples in the same vicinity (Fig 2). Despite a low relative abundance of those shared gSTs in environmental samples, linked to the high alpha diversity in these sample types, those shared gSTs were among the most frequently detected (Fig 4, S1 Fig in S2 File). These results show at an unprecedented resolution that invasive species like brushtail possums can be a source of *E. coli* in New Zealand waterways. In particular, the *E. coli* strain belonging to phylogroup B2 and associated with gST535 was present in many of our samples and core SNP analysis of WGS data confirmed the presence of closely related isolates (<10 SNPs) across sample types and time (Fig 3).

All water samples collected from the Mākirikiri Stream had *E. coli* concentrations exceeding the alert concentrations in the NZ national recreational water quality guidelines (260 MPN) of *E. coli*/100 mL [ref. 36], S2 Table in S1 File) and two measurements exceeded 540 MPN, the concentration at which waterways are considered unsuitable for recreation in New Zealand. Although the number of samples in this study and in previous work [34] was not sufficient to provide reasonable statistical power in testing for compliance with NZ long-term grading water quality standards (minimum of 60 samples over five years), they indicated that, at least at the time of sampling, there was evidence suggesting that the Mākirikiri Stream was exposed to fecal contamination and there was a potential public health risk associated with the two sites sampled. Additionally, the possum density appeared high in surrounding environs. Although not a robust estimate of abundance, the removal rate of possums over two months of trapping was roughly over 6 possums/ha (49 possums from 8 ha), as could be expected in a mixed broadleaf-podocarp forest edging pastures in the absence of control measures [15]. The rate of captures for other pest species such as rats was lower, potentially due to competition with possums [38] or aversion to the traps (neophobia), a frequent behavior with rats [39]. This is illustrated with the first ship rat capture happening almost a month after the first possum. The finding of the same clones of *E. coli* potentially circulating between animal and environmental sources supports evidence that possums, as well as other introduced predators and avian species may contribute to fecal contamination in waterways, as expected for a microorganism that spreads by the fecal-oral life cycle. Long-term monitoring of water quality where new possum control measures are undertaken would be needed to confirm if sustained pest control programs are followed by an improvement in water quality.

Using the DivNet method for diversity estimation, the community profiles of *Escherichia* determined by *gnd* metabarcoding revealed a significantly higher diversity of *Escherichia* gSTs in periphyton, water, and sediment samples compared to animal samples but we failed to find evidence for a difference in measured community composition between sample types. In addition, several gSTs were shared between animal and environmental samples (Fig 4, S1B Fig in S2 File). The same observations were noted when assessing diversity between *Escherichia* isolates obtained from both sample types. The higher alpha diversity in all environmental samples except soil could be expected, as water, periphyton and sediments are exposed to sources of *E. coli* spanning from the entire stream catchment to the sampling points, while soil samples represent a geographical point exposure within the localized sample sites.

The composition of *Escherichia* communities in different samples was assessed at the sample and sample type level using *gnd* metabarcoding and Illumina MiSeq sequencing, and at the isolate level using Sanger sequencing on up to two isolates per sample. These culture-independent and culture-dependent methods yielded valuable insights into the composition of *Escherichia* communities in the samples. Both sequencing methods target the same sequence of the *gnd* gene and use similar primers (with/without barcodes). The enrichment step used prior to metabarcoding and isolation may influence the growth of some strains over others which may

impact the results, notably in terms of abundance. Contrary to more ubiquitous multi-copy marker genes like 16S rRNA, the *gnd* marker gene is a single-copy gene not universally present and is thus theoretically found in lower concentration in samples (especially environmental samples where *Enterobacteriaceae* such as *E. coli* are not expected to grow). Previous attempts to extract sufficient *Escherichia* DNA for sequencing without the enrichment step had high rates of failure, especially for sediment and bird samples. Also, previous tests on calf feces had demonstrated that enrichment-based DNA extraction methods resulted in increased richness compared to direct DNA extraction, but the primary factor influencing the variation in richness remained the sampled animal [37]. An analogous longitudinal study of commensal *E. coli* strains in an Australian population of mountain brushtail possums (*Trichosurus cunninghami*) also showed in that species, without pre-enrichment, a very low culture-based alpha diversity and an average of 2.2 strains per possum, with changes in the main strain over sampling occasions [40]. One could also argue that the *Escherichia* species putatively favored by this enrichment step are those that would be detected by routine FIB enumeration methods which use enrichment such as Colilert and are, therefore, ultimately of greater interest.

Using cultures originating from complex fecal or environmental samples, WGS data has demonstrated the taxonomic diversity of *E. coli* phylotypes and closely related *Escherichia* species [8, 34]. However high throughput metabarcoding of amplified PCR products provides enhanced resolution over culture-based methods improving the detection of strains present at low-abundance. This is especially true when the within-host diversity of *Escherichia* is high, as has recently been described in wild species sharing the same habitat [8]. The *gnd* allele is located close to the O-antigen biosynthesis gene cluster, a highly recombinogenic area on the *E. coli* chromosome [41] and displays a high frequency of polymorphisms [42–47]. It has consequently been described as a passive hitchhiker of recombination events that determine antigenic changes of the lipopolysaccharide moiety [43]. The polymorphic nature of the *gnd* locus requires the use of degenerate primers [37] that may amplify some gSTs more efficiently than others during *gnd* PCR amplification and sequencing reactions, thus potentially influencing the results in terms of read abundance. Consequently, results were primarily interpreted in terms of presence/absence of a gST across all samples, and analyses were conducted with methods accounting for unobserved taxa [48]. With these caveats in mind, both methods led to similar conclusions with the most abundant isolates and reads shared between sample types despite a higher alpha diversity in environmental samples (Fig 4, S1, S2 Figs in S2 File).

gST535, which had the highest prevalence in environmental and animal samples (Table 3) was detected previously in the same geographic location [34], suggesting that this *E. coli* strain could be endemic to the area. WGS data from gST535 isolates indicated the presence of several adherence factors such as genes of the *sfa* fimbrial operon (S4 Table in S1 File) and Type 1 fimbriae (observed in all 101 isolates), present in uropathogenic *E. coli* (UPEC) [49], but also colicin and microcin genes (*cea*, *colE6*, *mchB*, *mchC*, *mchF*, *mcmA*) [50], suggesting that these virulence factors may offer advantages for survival in the environment [51, 52]. The surveillance of human cases of disease due to *Escherichia* in New Zealand is limited to Shiga toxin-producing *E. coli* (STEC), and MLST data is available since 2019, when WGS became the preferred source of typing (https://surv.esr.cri.nz/enteric_reference/vtec_isolates.php, last accessed 23/12/2022). Sequence types ST681 and ST11707, found in gST535 isolates in this study and previous work [34], were not found in human cases notified between 2019 and 2021 in New Zealand, suggesting this strain has limited public health implications. In support of this hypothesis, a phylogenetic tree of all internationally published ST681 and ST11707 available in Enterobase [26] is presented in S5 Fig in S2 File and indicated that New Zealand strains did not cluster with isolates recovered in other countries.

WGS data of other isolates also reflected a limited risk to public health. The absence of notable clinically relevant antimicrobial resistance genes was consistent with another prior study that did not identify extended-spectrum β-lactamase-producing *E. coli* in the Mākirikiri Stream [53]. The clonal group possessing an *eae* β2 allele found in five possums (phylogroup B2 O87:H6 ST3303) had previously been detected in a possum fecal sample in the reserve [34]. Another *E. coli* phylogroup D O7:H6 ST362 had also been isolated in avian feces and water near the Mangatera River (about 50m upstream of the 'Confluence' sampling site) and from the Mākākahi River (ca. 50 km south of the reserve) but this isolate lacked the *eae* α1 gene [34]. Although no *stx* (Shiga-toxin) genes were identified in any of the isolates which underwent WGS (S4 Table in S1 File), four STs detected in this study were also detected in human STEC cases in New Zealand; ST10 (corresponding to gST424, phylogroup A, found in periphyton, ship rat and avian feces samples), ST388 (gST248, B1, periphyton), ST357 (gST258, B2, ship rat, ferret and avian feces samples), and ST141 (gST258, B2, three possums gut contents). ST10 was the third most frequent cause of STEC infection and notified in 71 cases over the three reported years (https://surv.esr.cri.nz/enteric_reference/vtec_isolates.php, last accessed 23/12/2022). The other three STs were detected in a single case each. The serotype of the three ST10 found in this study (O26:H32) was unrelated to STEC serotype O26:H11, a widespread zoonotic pathogen found in cattle in New Zealand [phylogroup B1, ST21 or ST29, ref.54] and internationally [55, 56] and one of the 'top 7' STEC [54]. ST141 was the only ST for which serotypes were similar in this study and in human cases (O2:H6). ST141 was also detected in the aforementioned longitudinal study of commensal *E. coli* strains in mountain brushtail possums (*Trichosurus cunninghami*) using MLST and the Clermont quadruplex PCR [40]. That same study identified the most common *E. coli* strains belonged to phylogroup B2 (91%) and ST141. Comparative genomics of the ST141 from possums and the local STEC ST141 isolated from humans would provide further insights into any phylogenetic relationship, pathogen evolution and zoonotic transmission from possums. While the *Escherichia* isolated in this study were mostly not of public health concern *per se*, they may still be indicators of the presence of other waterborne pathogens, such as *Campylobacter* that have previously been identified in possums and wild birds [57, 58].

The *uidA* RT-PCR was indicative for the presence of putative cryptic clade isolates from 11/ 106 samples. The gSTs of all 17 *uidA*-negative isolates typed were identical to gSTs previously found in *E. marmotae*. WGS confirmed the species assignment underlining the usefulness of *gnd* sequence typing as a convenient typing method. These *E. marmotae* were isolated from a range of animal and physical habitats illustrating their wide distribution within this environment (Table 1). The same 'cryptic gSTs' were found with metabarcoding, with no obvious contrast of richness between animal and environmental samples (Table 1) and mostly in low abundance (in terms of reads), except for some birds and soil samples. Interestingly, despite accounting for 5.7% of all reads (5.4% of reads from an animal source), 'cryptic gSTs' reads (i.e., previously described only in cryptic clades) represented 28% of reads from avian feces. While the contribution of possums as a reservoir of *E. coli*-like naturalized *Escherichia* species appears limited in this study, the results from bird feces add to numerous reports associating cryptic clades and avian samples [30, 32, 59, 60] and warrant further investigation. Identification of birds as fecal reservoirs of cryptic clades of *Escherichia* would challenge the environmental hypothesis [31] where animals are suggested to represent a "spillover" host with only transient passage through the intestine.

In our previous work, the *gnd* metabarcoding method was used to examine *E. coli* populations from calf feces [37], but this work demonstrates that the method also allows the analysis of *Escherichia* populations in other environmental samples such as water, periphyton, soil and sediment. Despite the ability of the *gnd* primers used in metabarcoding to amplify *gnd* alleles

present in non-*Escherichia Enterobacteriaceae* such as *Enterobacter*, *Citrobacter*, *Serratia* and *Salmonella* [37], an overwhelming majority of reads were found to match the gndDb. This observation could be attributed to the type of samples under investigation, considering that these bacteria are not typically expected to be abundant in environmental samples. We employed an emerging approach for estimating alpha and beta diversity and assessing differences between sample types [48]. Unlike the widely debated methods of normalization and rarefaction [61, 62], the DivNet approach utilizes all ASVs and reads, accounts for unobserved "species" (here gSTs) and provides variance estimates [48, 63]. DivNet also works by comparing habitats (sample types) rather than individual samples, in contrast to other methods such as Principal Coordinates Analysis and permanova. It therefore provides a more accurate image and comparison of *Escherichia* communities in the different types of samples. As environmental and wildlife samples are under-represented in current *Escherichia* databases [9, 31], unknown types are more likely to be found in these sources than in human or livestock species, as illustrated by the isolation of two different MLST sequence types not described elsewhere: ST11707 found in water, soil, possum, rat, and avian feces samples in this study (S4 Table in S1 File) and prior work [34]; and ST14827 found in soil and possum in this study (S4 Table in S1 File).

In conclusion, this study provided valuable insights into the diversity and distribution of *Escherichia* populations at the wildlife-environment interface. By using a combination of methods at the sample and isolate level, we provided evidence that invasive species such as brushtail possums could be a source of *E. coli* in New Zealand waterways, with clonal strains of *E. coli* highly prevalent in both animal and environmental samples in the same vicinity. It is therefore realistic to hypothesize that the removal of invasive pests from the environment may also improve microbial water quality assessments in addition to enhancing endemic biodiversity. While *E. coli* strains isolated from fecal sources in this study had limited pathogenicity, public health implications can still arise from direct defecation or mobilization of fecal pathogens excreted by wildlife into waterways. This study also used high-resolution genomic methods for studying *Escherichia* populations in wild species/habitats, which can complement conventional fecal source tracking methods when unusual *E. coli* counts are detected [64] and which could also be adapted and applied to other ecological systems. Furthermore, the study highlighted the need to account for the impact of fecal sources from invasive animal species during public health risk assessments of waterway contamination.

## Materials and methods

This study was conducted following the Animal Welfare Act 1999 and the protocol was approved by the AgResearch Animal Ethics Committee (AE 15061). Approval for the work to occur in the Mākirikiri Reserve was provided by the Tararua District Council, Dannevirke.

### Study site and fieldwork

Animal and environmental samples were taken in the Mākirikiri Reserve, a 15-ha mixed podocarp-broadleaf forest and remnant of the Seventy Mile Bush on the outskirts of Dannevirke, North Island, New Zealand. Four different types of traps were deployed in the Reserve for the purposes of this study, and in association with pest control initiatives to improve endemic biodiversity. Mammals were captured using 16 Timms traps (Connovation Ltd, Auckland, NZ), 2 Trapinator traps (Cmi Ltd, Auckland, NZ), and 4 A12 traps (GoodNature Ltd., Wellington, NZ) mainly targeting possums, and 13 A24 traps (GoodNature Ltd., Wellington, NZ) mainly targeting other invasive species, namely rodents, hedgehogs, and small mustelids. These specific kill-traps were selected as the most appropriate for humane killing while allowing for a

reasonably good recovery rate and relatively intact carcasses for gut contents recovery and *E. coli* isolation (i.e., limited damage of the distal end of the gastrointestinal tract and intact gut contents). As with those traps animals are killed instantly upon capture, methods of anesthesia and analgesia and efforts to alleviate suffering were not applicable. Timms traps were baited with apple and a mix of cinnamon and sugar, Trapinator and A12s with a commercial Possum lure, and A24s with a commercial Rat Lure (GoodNature Ltd.). Traplines were established in the bush and along the Mākirikiri Stream, covering an area of approximately 8 ha (Fig 1). Traps were monitored at least once every three days. The number of captures per 100 trap-nights was calculated according to Cunningham and Moors [65]. Carcasses were collected over a period of two months, from early November 2020 to early January 2021. Dead animals were collected and stored at +4˚C before being brought to the Hopkirk Research Institute, Palmerston North.

## Sample collection

Samples were taken from (a) the gut contents of possums and other introduced predators (hedgehogs, rodents, and mustelids) recovered from the traps; (b) environmental samples (surface water during base flow conditions, sediment, soil and periphyton), collected from the Mākirikiri Stream ('Mākirikiri'), like in [53], and just upstream from the confluence of the Mākirikiri Stream with the Mangatera River ('Confluence', Fig 1), at the same sampling site as in [34]; and (c) avian/mammalian fecal samples found opportunistically in the environment during sampling visits (two avian and two mammalian samples per sampling site and visit). On average two days after collection (range 0–5 days), post-mortem sampling of gut contents was performed in a Class II biosafety cabinet to avoid sample contamination. Carcasses were sprayed with 70% ethanol before the abdominal cavity was opened, and about 5cm of the distal colon was aseptically removed and its contents emptied into a sterile collection tube. Environmental sampling visits occurred on 7 and 17 November and 4 and 18 December 2020, and all associated samples were processed on the same collection day. Surface water was sampled first, in bottles that were rinsed with sample water before collection and starting at the Confluence site to avoid downstream disturbances and contamination. Sediment samples (~100 g) were collected using a stainless-steel shovel and *ca*. 3 mm-mesh sieve, soil samples (~70 g) were collected using a sterile 150 mm stainless-steel corer, and periphyton by wiping a sterile sponge swab (EZ-Reach Sponge Sampler, World Bioproducts, WA, USA) on ~100 $cm^2$ of a fully submerged rock. Feces samples found 1–5 m around the environmental sampling point were collected either using a sterile container with a scoop cap or sterile Amies swab (Copan Diagnostics Inc., Brescia, Italy).

## Laboratory and data analyses

***E. coli* enumeration and enrichment.** Enumeration of coliforms and *E. coli* in water (100 mL) was undertaken using the Colilert-18 Quanti-Tray/2000 method (IDEXX, NZ). Enrichments were made for all sample types and incubated at 35˚C for 18–21 h. Water (100 mL) was filtered through a 0.45 μm nitrocellulose membrane filter which was then added to EC broth (10 mL, Oxoid, Hampshire, UK). For sediment and soil samples, 1 g was added to EC broth (9 mL). For periphyton samples, 35 ml of EC broth was added to the dry sponge swab and stomached for 60 s and 10 mL of the resulting mix was used. For feces and gut contents, the Amies swab or a weighed sample was mixed into EC broth (swab into 10 mL or 250 mg into 24.75 mL).

**Isolation of putative *E. coli* and isolate-level DNA extraction.** For each of the different sample types, 10 μL of enrichment was plated onto CHROMagar ECC plates (Paris, France)

and incubated overnight at 35˚C. After incubation, four well-spaced blue colonies were sub-cultured for purity onto MacConkey agar plates (Fort Richard, NZ) and incubated at 35˚C for 18h. For each of the four subcultured isolates, one or two well-spaced colonies were removed from the purity plate and resuspended in 400 μL Milli-Q water, briefly vortexed and incubated at 100˚C for 10 min before storage at -20˚C as a DNA template (boiled lysates) for subsequent PCR analyses. In addition, six to eight well-spaced colonies were removed from a purity plate, resuspended in a cryovial containing 1 mL Brain Heart Infusion broth (Fort Richard, NZ) and 450 μL glycerol (30% w/v), and stored at -80˚C as a glycerol stock isolate suspension.

***uidA* real-time PCR and confirmation of species.** A RT-PCR targeting the *E. coli-* and *Shigella*-specific β-glucuronidase gene *uidA* was conducted on boiled lysates for each isolate. Primers and probe used and RT-PCR conditions were as described in [66]. Reactions were performed for 35 cycles in a total volume of 20 μL consisting of 0.4 μM each of forward and reverse primers, 0.15 μM of probe, 4 μL of PerfeCTa qPCR ToughMix (Quantabio, DNAture, Gisborne, New Zealand), 12.1 μL of PCR grade water and 2 μL of isolate boiled lysate DNA template using a Qiagen Rotor-Gene Q machine (Bio-Strategy Ltd, Auckland, New Zealand). Reactions with a cycle threshold (Cq) $\geq$ 35 were deemed negative and associated isolates were putatively considered as cryptic clades.

***gnd* sequence typing of *Escherichia* isolates.** The *gnd* PCR was performed with KAPA HiFi HotStart ReadyMix in a total volume of 20 μL using the primers 2*gnd*F 5′–TCYATYATGCCWGGYGGVCAGAAAGAAG (*gnd* coordinates 415 to 442) and 2*gnd*R 5′–CATCAACCARGTAKTTACCSTCTTCATC (*gnd* coordinates 754 to 726) at a final concentra-tion of 0.3 μM and 1 μL of isolate boiled lysate as a template [37]. The PCR cycle consisted of a single denaturing step at 95˚C for 3 minutes followed by 30 cycles of 98˚C for 20 seconds, 63˚C for 30 seconds and 72˚C for 30 seconds and a final elongation step of 72˚C for 5 minutes followed by a 12˚C hold step using a T100 Thermal Cycler (Bio-Rad, Auckland, NZ). To ensure there was a major PCR product at approximately 340 bp, 2 μL of amplicons were run on a 2% (w/v) agarose gel stained with RedSafe (Custom Science, Auckland, NZ) for 40 min at 90 V, and subsequently visualized using UV illumination in a Gel Doc 1000 System (Bio-Rad Laboratories Inc., Hercules, CA, USA). Remaining volumes of amplicons were purified using the QIAquick PCR purification kit (Qiagen, Bio-Strategy, Auckland, NZ) and their DNA con-centration quantified by spectrophotometry using a NanoDrop spectrophotometer (Thermo-Fisher Scientific, Auckland, NZ). The DNA concentration in samples was diluted to 5 ng/μL with Milli-Q water and subsequently submitted for Sanger sequencing at Macrogen (Seoul, South Korea). The .ab1 files obtained were subsequently manually curated and aligned in Gen-eious Prime (Biomatters Ltd.) before gST assignment using the custom gndDb database ver-sion 20220902 [ref. 35].

**Whole genome sequencing and phylogenetic analyses of isolates.** DNA extractions and library preparations for WGS were undertaken on 101 purified isolate cultures as previously described [34]. Briefly, the QIAamp DNA minikit (Qiagen, Hilden, Germany) and Nextera XT DNA library preparation kit (Illumina, San Diego, CA) were used and WGS undertaken by Novogene Limited (Beijing, China) using the Illumina HiSeq X paired-end v4 platform (2 x 125 bp). *De novo* assembled genomes were obtained using the Nullarbor pipeline 2.0.20191013 [ref. 67]. It includes Trimmomatic 0.39 [ref. 68] to trim reads, SKESA 2.4.0 [ref. 69] for *de novo* assembly, Prokka 1.14.6 [ref. 70] for annotation, ABRicate 1.0.1 for virulome and resis-tome analysis (Seemann T, *ABRicate*, **Github** https://github.com/tseemann/abricate) using as databases respectively VFDB [2021-Mar-28] [ref. 71] and NCBI AMRFinderPlus (2021-Mar-27) [ref. 72], mlst 2.19.0 (Seemann T, mlst **Github** https://github.com/tseemann/mlst) for sequence type (ST) identification using the Achtman seven-locus multilocus sequence typing (MLST) scheme and PubMLST database [73], and snippy 4.6.0 and snp-dists 0.7.0 [ref. 74] for

core SNP phylogeny. The genome of *E. coli* AGR4059 (gST535), isolated from possum feces in the same reserve in March 2018 [ref. 34], was used as an internal reference to map sequence reads and generate SNPs from genome alignments. The draft assembled genomes were further characterized using ECTyper [75], the ClermonTyper web interface [76], and mash analysis [77, 78].

**Sample-level DNA extraction.**   For each sample, 1 mL of enrichment was centrifuged at 13,000 g for 1 min, the supernatant discarded, and the pellet resuspended in 1 mL of sterile phosphate buffered saline (0.1 M, pH 7.4). The pellet resuspension was then centrifuged again, the supernatant discarded, and the pellet resuspended in 1 mL of Milli-Q water and incubated at 100˚C for 10 min before storing the boiled lysate at -20˚C. In addition, 1 mL of enrichment from each sample was mixed with 450 μL glycerol (30% w/v) and stored at -80˚C.

***gnd* metabarcoding.**   Metabarcoding targeting the partial *gnd* allele amplicon generated from the boiled lysate from each separate enrichment was performed as described previously [37] with subsequent analysis of the Illumina MiSeq sequence reads using the packages *dada2* version 1.24.0 [ref. 79] and *phyloseq* version 1.42.0 [80] in R version 4.2.1 (2022-06-23). Equimolar amounts of separate purified PCR products with a unique combination of barcoded primers from six different *E. coli* were pooled for inclusion in the sequencing as a mock community positive control.

Reads were filtered by quality; forward reads were trimmed at position 240 and reverse reads at position 160, all reads were truncated at the first instance of a quality score below two and those with expected errors above two after truncation were discarded. Quality assured reads were then denoised and merged to produce ASVs; chimeric ASVs were then removed. Taxonomic assignment of *gnd* Sequence Types (gSTs) to 284 bp ASVs was done using the gndDb database version 20220902 [ref. 35] as the reference and the function *assignSpecies()* from *dada2* (exact match). Alpha and beta diversities were estimated using the *DivNet* package version 0.4.0 [ref. 48]. *DivNet* accounts for unobserved taxa (i.e., gSTs), under- and oversampling, and taxon-taxon interactions. Differences in alpha diversity between sample types were tested with the function *betta()* from package *breakaway* version 4.8.4 [ref. 63]. The null hypothesis of equal median measured relative abundance across sample types was tested with the *testBetaDiversity()* function from the *DivNet* package with a diagonal design matrix using a bootstrapped pseudo-F test (10,000 iterations). The ferret sample was not included in the beta diversity measurements as there was only one observation for this species. The upset plot was produced in R version 4.2.2 using packages *ggplot2* version 3.4.1 [81] and *ComplexUpset* version 1.3.5 (https://github.com/krassowski/complex-upset).

## Supporting information

**S1 File. Supplementary Tables S1-S5.**
(XLSX)

**S2 File. Supplementary Figures S1-S6.**
(PDF)

## Acknowledgments

We would like to thank mana whenua involved in the trap monitoring and carcass removal, Tararua District Council staff and Rose Collis (AgResearch) who helped transport the carcasses to Hopkirk Research Institute, and Lauren Gadd, a Pūhoro STEMM Academy summer intern who helped with the environmental sampling. We also thank Darren Peters (GoodNature) and Lisa Whittle (Ruahine Whio Collective) for their advice and help with trap

placement and James C. Russell (University of Auckland) and Bruce Warbuton (Manaaki Whenua—Landcare Research) for their advice on the choice of traps.

## Author Contributions

**Conceptualization:** Marie Moinet, Patrick Biggs, Jonathan Marshall, Richard Muirhead, Megan Devane, Rebecca Stott, Adrian Cookson.

**Data curation:** Marie Moinet, Patrick Biggs, Jonathan Marshall, Adrian Cookson.

**Formal analysis:** Marie Moinet, Patrick Biggs, Jonathan Marshall, Adrian Cookson.

**Funding acquisition:** Adrian Cookson.

**Investigation:** Marie Moinet, Lynn Rogers, Adrian Cookson.

**Methodology:** Marie Moinet, Patrick Biggs, Jonathan Marshall, Adrian Cookson.

**Project administration:** Adrian Cookson.

**Resources:** Marie Moinet, Adrian Cookson.

**Validation:** Patrick Biggs, Jonathan Marshall.

**Visualization:** Marie Moinet.

**Writing – original draft:** Marie Moinet.

**Writing – review & editing:** Marie Moinet, Lynn Rogers, Patrick Biggs, Jonathan Marshall, Richard Muirhead, Megan Devane, Rebecca Stott, Adrian Cookson.

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
