## [Decision Letter · Decision Letter 0]

24 Aug 2023

PONE-D-23-24161High-resolution genomic analysis to investigate the impact of the invasive brushtail possum (Trichosurus vulpecula) and other wildlife on water microbial quality assessmentsPLOS ONE

Dear Dr. Cookson,

Thank you for submitting your manuscript to PLOS ONE. After careful consideration, we feel that it has merit but does not fully meet PLOS ONE’s publication criteria as it currently stands. Therefore, we invite you to submit a revised version of the manuscript that addresses the points raised during the review process.

Two expert reviewers evaluated your manuscript and both suggested minor revisions. They both provided valuable comments to improve the quality of the manuscript. I suggest that you go through the comments carefully and address each of them. Where corrections cannot be made, you should kindly provide a valid rebuttal.

We look forward to receiving your revised manuscript.

Kind regards,

Christopher Adenyo, Ph.D.

Academic Editor

PLOS ONE

Journal Requirements:

Reviewers' comments:

Reviewer's Responses to Questions

**Comments to the Author**

1. Is the manuscript technically sound, and do the data support the conclusions?

Reviewer #1: Yes

Reviewer #2: Partly

2. Has the statistical analysis been performed appropriately and rigorously? 

Reviewer #1: Yes

Reviewer #2: Yes

3. Have the authors made all data underlying the findings in their manuscript fully available?

Reviewer #1: Yes

Reviewer #2: Yes

4. Is the manuscript presented in an intelligible fashion and written in standard English?

Reviewer #1: Yes

Reviewer #2: Yes

5. Review Comments to the Author

Reviewer #1: The presented study focuses on the characterization of Escherichia populations in fecal specimens from possums, specifically exploring the potential contribution of this invasive predator species to fecal E. coli bacteria and other E. coli-like naturalized Escherichia species that could impact water quality assessments. The research methodology involves applying advanced techniques such as whole genome sequencing and phylogenomic analysis to isolate and characterize E. coli strains present in possum fecal samples. Additionally, the study employs amplicon metabarcoding targeting the gnd gene to directly profile the E. coli community in these samples. This approach is based on comparing gnd amplicons with a custom database containing distinct gnd sequence types (gSTs) found in E. coli and E. coli-like species.

The whole genome and metabarcoding approach using amplicon sequencing of the gnd gene provided a broader view of the E. coli community composition across diverse sample types. The results indicated the prevalence of specific gSTs, such as gST535 and gST258, across various sources. Furthermore, the study identified a range of 'cryptic gSTs' associated with cryptic clades, some of which were previously linked to E. marmotae, E. ruysiae, and E. whittamii.

This research not only addresses a critical environmental concern but also advances our understanding of the complexities within microbial communities and their impact on water quality assessments. By combining cutting-edge genetic techniques and a comprehensive sampling strategy, the study offers valuable information that can inform water management strategies and conservation efforts, potentially leading to improved water quality and ecosystem health. I therefore recommend acceptance. However, I would like the authors to attend to the following minor comments below.

Line 62-63. Please insert “an”. Considerable resources are spent in an attempt to control possum

Line 123: The “of” sentence should be changed to “as”.

Line 252: Since alpha diversity is used to established the diversity within a sample, I am thinking that the statement “The difference in alpha diversity between water and other sample types was significant at the 0.05 level, except for soil (p = 0.789) and ship rats (p = 0.074, Table 2)” is not reflecting the actual meaning and should be rephrased. This is a suggestion for your consideration “The difference in the alpha diversity of water and other samples types was significant at 0.05 level, except for soil (p = 0.789) and ship rats (p = 0.074, Table 2).”

Line261: As you indicated in parenthesis some gSTs were previously found in E. marmotae an E. ruysiae and E. whittamii, I kindly want to find out if these works have been published? If yes kindly included the reference.

Line 281: Please rephrase. I suggest you make it “two measurements exceeded 540 MPN”

Line 290: Please the unit, I think you should state it as 6 possum/ha for better clarity.

Line 347: Please insert “that”. “Suggesting that these virulence factors may offer advantages for survival”

Line 351: Please are you suggesting that ST681 and ST11707 are associated with gST535? If that is the case, I will suggest that you rephrase the sentence for better clarity. “Sequence types (ST681 and ST11707) associated with gST535 were not found in human cases notified between 2019 and 2021 in New Zealand, suggesting this strain has limited public health implications” for your consideration. Additionally, I would like the author to provide further explanation in their materials and method section how they determined the association between the ST and gST.

Line 355: Please insert “that” after the word indicated.

Reviewer #2: Overall, the study conducted by Moinet et al, uses appropriate methods and analyses that provide valuable information on the E. coli strains found in an invasive wildlife species; the common brushtail possum. Perhaps unsurprisingly, the study provides evidence that E. coli from common brushtail possums contributes to the E.coli found in the environment (including water) where the possums are found. However, there is the potential for the authors to go further in describing and discussing what proportion of total E. coli levels in the environment possum E. coli constitutes. The authors adequately discuss the limitation of the sampling methods used.

Below I have provided specific line by line comments and suggestions, including a query regarding the analysis for differences in community composition between sample types.

Abstract

L30-32: It would be good to include some more specific details of sharing between possums and environmental samples in the abstract. What proportion of environmental isolates were also detected in possums? What proportion of possum isolates were shared with other animals?

Results

L128: please provide a brief description of the culture based methods used.

L142: clarify that the 55% refers to the two gSTs combined.

L146-149: So the most abundant gSTs in possums were not detected at high rates in the environment and other animals but were present. A version of this would be good to include in the abstract.

L170: why was the Clermont method of E. coli phylogenetic grouping not conducted on all isolates to give a more representative picture of the phylogroup distributions between the sample types? This could easily be done on the 420 isolates – well at least those uidA positive.

L172: How did these correspond to the gSTs?

L250-251: Please present the statistical significance of these comparisons.

Discussion

L273: Prevalence yes, but these gSTs did not appear to be at high abundance in the environmental samples according to this statement from the results: “The three most frequent gSTs were gST152 (five isolates, 7.8% of environmental 145 isolates), gST535 and gST587 (both four isolates, each 6.3% of environmental isolates respectively).”

L298: suggest changing “confirm” to “show” or similar. At present, this study does not provide any evidence that removing possums would increase water quality. Could the authors do an analysis to assess this? If you assume all gST535 and maybe gST258 isolates in the water were from possums, then how much would the E.coli concentrations in the water be reduced if they were removed?

L302: Which analysis showed that there was no difference in community composition between sample types? This seems surprising and I’d like to see a permanova test to confirm that as well as a PCoA of the Bray Curtis distances.

L229-340: This is a good point and I think any abundance measures from this data should be interpreted with caution. However, I do think that they have some value where there are big differences between sample types, which might be worth discussing with the appropriate caveats.

L353: But you just said monitoring is limited, so this statement can not be made with certainty.

L421: I think this is overstated. There needs to be evidence that possum gSTs are contributing to high E.coli loads in water before this statement can be made.

6. PLOS authors have the option to publish the peer review history of their article (what does this mean?). If published, this will include your full peer review and any attached files.

Reviewer #1: **Yes: **Justice Opare Odoi

Reviewer #2: No

---

## [Author Response · Author response to Decision Letter 0]

18 Oct 2023

All editor and reviewers comments and requests have been responded in the attached Response to reviewers

---

## [Editor Report · Decision Letter 1]

22 Nov 2023

High-resolution genomic analysis to investigate the impact of the invasive brushtail possum (*Trichosurus vulpecula *) and other wildlife on microbial water quality assessments

PONE-D-23-24161R1

Dear Dr. Cookson,

We’re pleased to inform you that your manuscript has been judged scientifically suitable for publication and will be formally accepted for publication once it meets all outstanding technical requirements.

Kind regards,

Christopher Adenyo, Ph.D.

Academic Editor

PLOS ONE
---

## [Editor Report · Acceptance letter]

8 Jan 2024

PONE-D-23-24161R1 

PLOS ONE

Dear Dr. Cookson, 

I'm pleased to inform you that your manuscript has been deemed suitable for publication in PLOS ONE. Congratulations! Your manuscript is now being handed over to our production team.

Kind regards, 

on behalf of

Dr. Christopher Adenyo 

Academic Editor

PLOS ONE